# Towards Building a Visual Behaviour Analysis Pipeline for Suicide Detection and Prevention

**DOI:** 10.3390/s22124488

**Published:** 2022-06-14

**Authors:** Xun Li, Sandersan Onie, Morgan Liang, Mark Larsen, Arcot Sowmya

**Affiliations:** 1School of Computer Science and Engineering, University of New South Wales, Kensington, NSW 2052, Australia; xun.li1@unsw.edu.au (X.L.); morganmliang@gmail.com (M.L.); 2Black Dog Institute, University of New South Wales, Randwick, NSW 2031, Australia; s.onie@blackdog.org.au (S.O.); mark.larsen@blackdog.org.au (M.L.)

**Keywords:** behaviour recognition, pedestrian tracking, pose estimation, suicide prevention, video surveillance

## Abstract

Understanding human behaviours through video analysis has seen significant research progress in recent years with the advancement of deep learning. This topic is of great importance to the next generation of intelligent visual surveillance systems which are capable of real-time detection and analysis of human behaviours. One important application is to automatically monitor and detect individuals who are in crisis at suicide hotspots to facilitate early intervention and prevention. However, there is still a significant gap between research in human action recognition and visual video processing in general, and their application to monitor hotspots for suicide prevention. While complex backgrounds, non-rigid movements of pedestrians and limitations of surveillance cameras and multi-task requirements for a surveillance system all pose challenges to the development of such systems, a further challenge is the detection of crisis behaviours before a suicide attempt is made, and there is a paucity of datasets in this area due to privacy and confidentiality issues. Most relevant research only applies to detecting suicides such as hangings or jumps from bridges, providing no potential for early prevention. In this research, these problems are addressed by proposing a new modular design for an intelligent visual processing pipeline that is capable of pedestrian detection, tracking, pose estimation and recognition of both normal actions and high risk behavioural cues that are important indicators of a suicide attempt. Specifically, based on the key finding that human body gestures can be used for the detection of social signals that potentially precede a suicide attempt, a new 2D skeleton-based action recognition algorithm is proposed. By using a two-branch network that takes advantage of three types of skeleton-based features extracted from a sequence of frames and a stacked LSTM structure, the model predicts the action label at each time step. It achieved good performance on both the public dataset JHMDB and a smaller private CCTV footage collection on action recognition. Moreover, a logical layer, which uses knowledge from a human coding study to recognise pre-suicide behaviour indicators, has been built on top of the action recognition module to compensate for the small dataset size. It enables complex behaviour patterns to be recognised even from smaller datasets. The whole pipeline has been tested in a real-world application of suicide prevention using simulated footage from a surveillance system installed at a suicide hotspot, and preliminary results confirm its effectiveness at capturing crisis behaviour indicators for early detection and prevention of suicide.

## 1. Introduction

Understanding of human behaviours through video analysis has boomed in recent years with the advances in deep learning and the ubiquity of cameras. There is growing interest in applying it to visual surveillance systems [1,2,3]. They have been widely deployed in both public and private locations, such as transport systems, parks, shopping centres, hospitals and prisons. Typical applications include access control, human identification, flow counting and detection of unusual or abnormal behaviours. Research in this area attempts to detect, track and analyse pedestrian behaviours to replace labour-intensive manual monitoring. However, there are several challenges to overcome to make such real-world applications a reality. Surveillance cameras’ angles, depth and resolution are usually difficult to control to fully capture non-rigid articulated movements of human bodies. Complex background noise and variable illumination conditions further add to the challenges of algorithmic analysis. Furthermore, while many researchers in the computer vision community aim to develop end-to-end action detection and recognition systems, surveillance applications often require a far more comprehensive approach that is not only capable of a single task, but also can provide information that may be used in multiple applications, such as instance searching of specific targets, flow counting, trajectory analysis and action recognition for specific scenarios. In this paper, we focus on this last application while trying to maintain the potential for other tasks. Finally, in the domain of abnormal behaviour detection, the paucity of data due to rare cases and privacy and confidentiality concerns pose further challenges. To date, the majority of the relevant research has been focussed on detecting certain incidents when they have already taken place and targets are known [4]. A more attractive and challenging approach for crisis detection is to automatically perform human action recognition and behaviour interpretation to detect early cues so as to identify persons in crisis and perform early interventions.

Suicide prevention and risk detection have long constituted an important area of study in psychiatry and public health [5]. Machine learning techniques have been introduced into the field via natural language processing techniques [6,7,8] for analysing text data posted on social media for the detection of suicide intentions. CCTV cameras may be installed at known suicide hotspots, which are often specific public sites that provide the means and are associated with a high frequency of suicide [9]. Currently, such systems are either utilised only with manual observation or integrated with basic visual processing tools to trigger emergency responses when a suicide attempt takes place, such as detecting individuals on the wrong side of a safety fence. However, both methods can be too late to prevent the tragedy, and manual observation is time and labour consuming. Suicide prevention researchers are working on identifying behaviours that precede suicide attempts. Detection of such behavioural cues will allow for early intervention. A major step would be to combine such information with machine learning technology so that pre-suicide behaviours can be captured automatically [10]. However, such multi-disciplinary approaches have not yet been reported in the literature to date.

In this study, we aimed to address this gap by incorporating observations from suicide prevention researchers to assist in the supervised learning of a visual action recognition algorithm—in particular, to identify pre-suicide indicators, to enable persons in crisis to be automatically identified from fairly small datasets. Specifically, a modular, human-centric visual pipeline is proposed for surveillance systems to separately provide multi-task functionalities for pedestrian detection, tracking, pose estimation and action recognition. In particular, due to the virtues of skeleton-based features and the key finding that human body gestures can be used for the detection of social signals that potentially lead to suicide attempts [11,12], a new 2D skeleton-based action recognition algorithm was proposed and used in the pipeline.

The contributions of this work are three fold: (i) a new visual behaviour analysis pipeline for an intelligent surveillance system is introduced; (ii) a new 2D skeleton-based action recognition model is proposed, which achieves good performance on both public datasets and private CCTV footage; (iii) an action recognition model has been trained for detecting specific pre-suicide behaviours identified by collaborating suicide prevention researchers. The whole pipeline has been tested using simulated footage from a surveillance system installed at a suicide hotspot. Preliminary results, which show the effectiveness of this system at capturing crisis behaviour indicators for suicide prevention, are provided in this pioneering research effort.

## 2. Background and Related Work

### 2.1. Intelligent Surveillance for Suicide Detection and Prevention

Researchers have been working on intelligent surveillance systems for many years, even before deep learning started dominating the area of video analysis. Revathi et al. [1] depicted such a system in four conceptual layers: surroundings modelling, which detects moving objects in the scene; object representation and classification; object tracking; and behaviour understanding. A similar framework can be found elsewhere [13]. Today, the first two problems can be jointly solved with a highly efficient and accurate single-shot object detector. Meanwhile, deep learning-based multiple-object tracking has also seen fast advances through the introduction of deep appearance embedding into the data association process. In this work, the framework is redefined into three conceptual layers: pedestrian detection, tracking and behaviour understanding.

Intelligent surveillance systems capable of action recognition have attracted growing attention in the past decade. Kim et al. [14] used skeleton-based action recognition to recognise the behaviour of persons handling objects such as a phone, a cup or a plastic bag in visual surveillance systems. Li et al. [2] proposed a novel group-skeleton-based human action recognition method for complex events. However, research on CCTV video analysis for suicide detection and prevention is very limited. Most available research only focusses on detection after a suicide attempt has already been made. Lee et al. [15] described a CCTV monitoring system to detect individuals who have already jumped from a bridge in Seoul. Bouachir et al. [16] developed a video surveillance system to detect suicide attempts by hanging. RGB-D cameras were used to detect 3D positions of body joints and hand-crafted features were extracted based on the joint positions. More specifically, the authors used pair-wise joint distances within each frame and between two consecutive frames as feature vectors for classification. A linear discriminant classifier (LDC) was trained to classify two groups of actions: non-suspicious actions (such as move, sit, wear clothes) and suicide by hanging. However, the training dataset was based on simulated video recordings taken in a room environment, and the classifier was designed to discriminate only one action from the rest of human actions. Therefore, the applicability of such a system is limited. Other similar studies have been performed also [17].

### 2.2. Pre-Suicidal Behaviour Recognition for Early Detection and Prevention

Studies [16], including from this group [12], have identified that the ability to prevent suicide is limited in current systems, mainly because an alert is only triggered when an attempt has taken place (e.g., when an individual has stepped onto railway tracks). The next step is to find ways to reach out to an individual prior to an attempt. One way is to identify patterns of behaviour preceding an attempt. One study [10] identified several behaviours by analysing 5-min segments of video footage from metro stations, with or without a suicide attempt at the end, and discovered that two behaviours (pacing back and forth from the edge of the platform, and leaving belongings on the platform) could identify 24% of the incidents with no false positives. Another study [18] identified five main behaviours preceding suicides at underground railway stations: station hopping and platform switching; limited contact with people; allowing trains to pass by; location where they stood on the platform; and repetitive behaviours, such as walking up and down the length of the platform.

Our study investigated whether these pre-suicidal behaviours can be detected automatically. One study that was most similar to the current work was by Reid et al. [11], in which the authors attempted to extract head poses to detect social signals for suicide prevention. They propose that the head pose reflects social signals that can be used to indicate where an individual’s attention is focussed. Head pose estimation was implemented using histogram of oriented gradients (HOG) with support vector machines. Both approaches use the human body for detection of social signals that potentially precede a suicide attempt and enable earlier crisis detection. In this work, more comprehensive information from both space and time domains is used, along with the skeleton movement of the whole body as a sequential feature series, to detect potential pre-suicide behaviours.

### 2.3. Skeleton-Based Action Recognition

Human action recognition has been an important and challenging area for computer vision. Extracting spatial-temporal feature representations directly from raw videos with deep-learning-based models such as two-stream approaches [19,20], 3D convolutions [21,22], LSTM methods [23] and RNN-based methods [24,25] are the most common ones. Compared to these methods that use raw RGB input to extract low level features for action recognition, skeleton-based action recognition that uses human joint information as input has been attracting more attention recently. Compared to heavy computation for the raw RGB and optical flow, computing a skeleton sequence is lightweight. Moreover, it does not contain colour information, and thus is not affected by the limitations of raw RGB inputs, including background clutter, illumination changes and appearance variations. It is therefore suited for surveillance camera systems installed in outdoor environments. Moreover, its exceptional capability to detect actions involving certain human postures is likely to bring great benefit to detection of human behaviour risk indicators in early suicide detection.

Based on the analysed strengths, skeleton-based action recognition was chosen in this work as the main function module. While many works use 3D skeleton information provided by RGB-D cameras or wearable devices, this work focusses on 2D skeleton-based action recognition in single RGB camera video, which is commonplace for many surveillance systems. Pose estimation algorithms provide skeleton information in 2D. A detailed review of skeleton-based action recognition from two perspectives is now provided: feature extraction and network structures.

#### 2.3.1. Feature Extraction for Skeleton-Based Action Recognition

Compared with hand-crafted features directly extracted from images or videos, skeleton information, also known as pose, is regarded as a high-level feature set, as it does not rely on appearance or colour information, and is more robust and discriminative in challenging real-world environments.

Researchers realised the importance of skeleton features for capturing human motion long before skeleton-based action recognition systems began to appear. Johansson [26] gave empirical and theoretical bases for key joints that can provide highly effective information about human motion. Yao et al. [27] pointed outed that pose features outperform appearance features. Jhuang et al.’s work [28] was one of the first to quantitatively compare the importance and significance of body joints for action recognition to that of other low/middle level features, such as histogram of oriented gradients (HOG) and motion boundary histograms (MBH) extracted directly from videos. The authors used dense trajectory [29] as a baseline and different low–mid–high level features to compare their performance. They observed that high-level pose features outperformed other features in action recognition. Moreover, good pose features need to consider not only the positions of joints, but also relations of joints and number of frames.

In general, skeleton features can be categorised into three groups: static Cartesian features, such as joint coordinates; geometric features to represent their geometric relationship, such as distances and angles; and motion features, such as velocity and acceleration. Researchers usually choose some of these features in their studies. Chen et al. [30] encoded both static features, including joint coordinates and geometric relational features, and motion features into a feature descriptor, namely, the geometric pose descriptor. They then used a semi-supervised distance metric learning to enforce the condition that similar poses be closer than unrelated poses. Their experimental results showed that such a descriptor effectively encodes human pose and captures temporal motion information. Yang et al. [31] proposed a hand-crafted location-viewpoint-invariant feature (namely, JCD) which computes Euclidean distances between pairs of joints, and used it together with two scales of scale invariant motion features as input, along with a CNN formed mainly by 1D ConvNet layers for processing. It has a simple and elegant design and is currently the state of the art (SOTA) on two public datasets: JHMDB [28] and SHREC [32]. However, further experiments on real-world applications are still necessary because of strict requirements for accurate and sufficient pose information.

#### 2.3.2. Deep Neural Network Architecture for Skeleton-Based Action Recognition

Researchers have used different deep neural network architectures to capture spatio-temporal information for skeleton-based action recognition. Here the focus is on, but not be limited to, 2D skeleton analysis.

A vast proportion of computer vision systems have benefited from using a CNN architecture due to its extremely powerful feature extraction that is translation-invariant and locally connected. However, while such characteristics are suitable for processing image-like data, they are insufficient to capture the spatio-temporal information in sequential data. One natural idea is to encode such temporal information into an image style format for CNN models to process. Choutas et al. [33] aggregated the heatmaps generated by the pose estimation process and encoded the whole clip into a fixed-size image-like representation with different colours representing different times. SOTA performance on JHMDB (70.8%) was achieved only by overly complex stacking of three different aggregation schemes; each one reached an accuracy of around 60%. A similar and more advanced approach has been proposed recently [34]. The method stacks the heatmaps of the entire video and uses an arbitrary 3D-CNN for 3D heatmap volume classification. We believe such approaches may not be suitable for real-time applications, as processing on ongoing video recordings will be necessary, rather than only focussing on classification of known video clips that allow for fixed heatmap aggregation. Ludl et al. [35] encoded skeleton joints from a sequence of frames (32 was used in the paper) into a new format, namely, encoded human pose image, where normalised joint positions were projected into RGB space. A shallow CNN was then used for classification based on such pseudo-images. This was a study very similar to the current one, in that they too used a modular design for real-world surveillance applications. Its features, however, only included Cartesian joint locations. While CNN filters are effective at capturing short-range patterns within local spatiotemporal regions, they simply cannot model space-time dependencies that extend beyond their small receptive fields. This is the common limitation of CNNs, which are strong at extracting visual features, but are not able to capture the dependencies in between.

RNNs and LSTM structures are able to capture relatively long dependencies by retaining useful information from all the steps ‘in memory’ via their ‘gates’. One of first papers using RNN style structures, and LSTM more specifically, in the domain of action recognition, was written by Donahue et al. [24]. They proposed an LSTM network that used a CNN to extract a fixed-length feature representation and then pass it into a recurrent-sequence learning model to process sequential video data. Their experiment shows that such an architecture has distinct advantages over traditional models that perform simple temporal averaging for sequential processing, due to its ability to model complex temporal dynamics. Instead of learning features from a CNN structure, high-level skeleton features have been used as input for RNN-style networks in recent years. Many studies have focussed on using joint coordinates as inputs and employing intentionally and carefully designed LSTM models, such as hierarchical architectures [36], part-based memory cells [37] and new mechanisms of regularisation and dropout [38]. Zhang et al. [39] made two main contributions: firstly, they introduced geometric features into LSTM networks for 3D skeleton-based action recognition. They observed that geometric features are more discriminative than joint location features. Secondly, compared to learning features with advanced models, they showed that defining good hand-crafted skeleton features for a basic model can be superior. The network design proposed in this paper was inspired by these observations. The main difference between the current approach and previous work lies in the use of two simple LSTM branches to explore the information provided by 2D geometric and non-geometric pose features, respectively. A final score fusion is used to combine the results to achieve better accuracy. The overall design is simple yet effective.

## 3. Methodology

A visual behaviour analysis pipeline for surveillance applications is proposed in this study, as shown in Figure 1. The intended application is to analyse human behaviours in the monitored area with a focus on early recognition of crisis behaviours preceding suicide attempts.

A modular pipeline is proposed. Each module is separately built in a top-down fashion, which means that it is not an end-to-end solution for action detection and classification. Each step in this pipeline uses the output from the previous module. For instance, the pedestrian detection module acts on the raw video footage, and its output is the detected bounding boxes. The pedestrian tracking module assigns a unique ID to each bounding box. The pose estimation module extracts human skeleton information from each bounding-box-enclosed region. All these three modules adopt well-established algorithms. For the action recognition module, a novel method is proposed that uses sequential skeleton information from the same pedestrian to classify and recognise actions of interest. A final module, namely, the logical layer, is built on top of the previous processing modules, using outputs mainly from the action recognition module and tracking module to recognise pre-suicide crisis behaviours.

The modular design is intended to enable each module to function independently depending on different user requirements. For a surveillance camera system, pedestrian detection output is often used to perform flow analysis, tracking output to build trajectories, and trajectory analysis for a variety of applications [40]. Moreover, such progressively abstract layers allow each component to be tested on various algorithms and switched on as needed. Meanwhile, machine-learning-based models for each task can be trained specially on private data for better adaptation to the intended application, such as customised detectors and action recognition models. The pipeline can be easily generalised to other scenarios just by using certain function modalities independently, or retaining partial functions in degraded conditions. For instance, when pedestrians are too far away for pose estimation, the detection and tracking modules can still provide useful information.

### 3.1. Pedestrian Detection, Pedestrian Tracking and Pose Estimation

For pedestrian detection which is the first module of the proposed pipeline, YOLOv5x [41] is used, mainly due to its suitable design for surveillance applications. It provides a good balance between accuracy and inference speed, and has adjustable model size for different deployment requirements. In addition, the mosaic augmentation and adaptive anchor mechanism that automatically adjusts the anchor size enables training of datasets from different camera views with ease. For training the pedestrian detection module for the application CCTV site, 1329 images from 32 short RGB clips with 1 frame/second rate and resolution of 704 by 480 were collected from both daily routine footage and the simulated footage that covers the main camera view. They also included various lighting conditions from early morning to late afternoon (thermal-based analysis during the hours of darkness will be discussed separately in a future study). Pre-trained weights from MS COCO [42] were used, and transfer learning was performed on the private CCTV dataset with a batch size of 16 and a total of 80 epochs. This model achieved 80.7% precision, 97.3% recall and mAP@.5 at 98.1% on this dataset. The speed of inference on average was 2.6 ms on a single GPU.

Tracking plays an essential role in surveillance camera systems. DeepSORT [43], a classical multi-object tracking (MOT) algorithm that benefits from outstanding design of deep appearance embedding and cascade matching, is adopted in the tracking module with modifications. First, the original detector Faster-RCNN is replaced by YOLOv5. DeepSORT uses a Kalman Filter framework with constant velocity motion and linear observation model: (*x*,*y*,α,*h*,vx,vy,vα,vh). The velocity information (vx,vy) is extracted and used as part of the motion feature that describes the movement of the overall bounding box at the action recognition stage. In addition, a basic API is built on top of the tracking algorithm to perform two additional critical functions for potential suicide prevention purposes: grouping and trajectory analysis. Grouping is a functionality to differentiate between a group of visitors and single visitor. A discovery from previous work [12] is that individuals in crisis are more likely to be alone rather than in a group. Therefore, visual analysis is used to classify whether a pedestrian belongs to a group of visitors or not. Three similarity criteria are used for grouping: distance, velocity and scale. Distance is defined by the Euclidean distance between every two human centres in each frame. Velocity is defined by the pixel movement between consecutive frames, calculated by bounding box velocity provided by the Kalman filter in DeepSORT. Pedestrians who stay close and at similar velocities have a good chance of belonging to the same group. In addition, scale is taken into consideration: scale similarity is calculated by the width and height of the bounding boxes. A similarity matrix is calculated for every two pedestrians for each of the three criteria, and a group is formed when all three costs for any two pedestrians are within a certain threshold, which is subject to different views. Then, any two-pedestrian groups with the same pedestrians are combined to form a larger group until no group has a common element. A second functionality for this API is trajectory analysis. The trajectory for each pedestrian is stored for a pre-defined duration. Since one of the potential high risk indicators is walking back and forth near the fence, the trajectory will be able to provide crucial information for early warning. For instance, in Figure 2, the trajectories are recorded for a duration of 2 s. Note that these two functions are essential building blocks that will be used in future research on trajectory based suicide detection.

Human pose estimation (HPE) is the problem of localisation of human joints in images and videos. When applying HPE to surveillance camera systems, low resolution and outdoor scenarios pose further challenges to accurate joint localisation. Therefore, the choice of the HPE method was focussed on selecting a more accurate and spatially more precise algorithm. In addition, since a top-down approach is adopted in the pipeline where an individual pedestrian is first detected and then the detected region used for pose estimation, multi-person pose estimation algorithms [44] and bottom-up approaches [45] were not considered. In this research, HRNet [46] was used due to its exceptionally good performance in the precise localisation of human joints. The key innovation of this work is that the network maintains a high-resolution representation branch throughout the whole process. Repeated multi-scale fusion is also introduced at different stages, which leads to rich high-resolution and semantically strong representation for the final prediction. The weights pre-trained on MSCOCO [42] and MPII dataset [47] are used. The module uses the human bounding box from the detection and tracking modules as inputs and estimates human joint locations in the cropped region.

### 3.2. Action Recognition

The last but most important component of the visual pipeline comprises a novel main function model for skeleton-based action recognition, and a new logical layer built based on expert knowledge from suicide prevention research and two model outputs: tracking and action recognition.

After pose information has been estimated, various skeleton features are extracted, as discussed previously: static, motion and geometric features. A custom-designed two-branch network is proposed that takes advantage of all three types of features extracted from the skeleton in a sequence of frames with a stacked LSTM structure. More specifically, static and motion features are fed into branch A, and the corresponding geometric features are fed into branch B. The two branches have the same structure but different hyperparameters tuned on different datasets. After each branch computes a softmax score on action classes, an element-wise multiplication score fusion is acquired to compute the final score for action prediction of a given sequence. An overview of the architecture is shown in Figure 3. In the following subsections, the feature extraction step and the main neural network components are introduced. Then, the target application with detailed design of the logical layer is presented.

#### 3.2.1. Feature Extraction from Skeleton Sequence

The output of the pose estimation module provides the joint coordinates in each frame. The coordinates are normalised in the following steps: first, divide the coordinates by the image width; then, retain 14 joints out of 18 coordinates by removing two eyes and two ears, since facial expressions are not under consideration; convert the image coordinates into a body coordinate system by setting the ‘neck point’ as the origin and transforming every other point relative to it; and finally, divide every value by body height (defined as distance between neck and thigh: middle point calculated by two thighs). Missing joints are filled by their corresponding points in the previous frame.

Then, three types of features (as reviewed in Section 2.3.1) are extracted: static, motion and geometric. Static features are normalised joint coordinates and consist of 28 dimensions. Motion features consist of three parts: body velocity calculated by subtracting adjacent frame positions of the neck joint; joint velocity calculated by subtracting adjacent frame positions of every joint available; and tracked bounding box velocity provided by the Kalman Filter in the tracking module, which is normalised by dividing the image width and body height. To increase the importance of body and tracked bounding box velocity, as they both represent movement of the whole body, they are given higher weights than the other features (10 times in the experiments). The overall dimension count for the motion feature is 68. Concatenating the 28 static features with the 68 motion features creates a 96-dimension feature vector. The reasons for this feature combination will be discussed in experiment A.

Geometric features are also referred to as relational features, as they describe the geometric relationships between joints. They are view-point invariant and serve as good complements to the static and motion features. Moreover, geometric features contain rich spatial information, and learning their variations over time will further improve the model’s capability to capture spatio-temporal information for action recognition. Five types of features are defined, as listed and described in Table 1. It is noted that although the feature normalisation and geometric features are meant to mitigate the influence of viewing angle, for best performance it is still recommended to retrain the model if a new camera with very different angle/distance is added to the analysis.

#### 3.2.2. Neural Networks

The proposed deep learning network for action recognition is based on the LSTM architecture. It is a modified version of RNN that is able to overcome the vanishing gradient problem and mitigate its incapacity with long-term dependencies [48], benefitting from LSTM’s three gates: forget gate $f_{t}$, input gate $i_{t}$ and output gate $o_{t}$:(1)$$\begin{array}{c} f_{t}=\sigma \left(W_{f}\left[h_{t-1},x_{t}\right]+b_{f}\right) \end{array}$$(2)$$\begin{array}{c} i_{t}=\sigma \left(W_{i}\left[h_{t-1},x_{t}\right]+b_{i}\right) \end{array}$$(3)$$\begin{array}{c} o_{t}=\sigma \left(W_{o}\left[h_{t-1},x_{t}\right]+b_{o}\right) \end{array}$$(4)$$\begin{array}{c} {\tilde{c}}_{t}=tanh\left(W_{c}\left[h_{t-1},x_{t}\right]+b_{c}\right) \end{array}$$(5)$$\begin{array}{c} c_{t}=f_{t}*c_{t-1}+i_{t}*{\tilde{c}}_{t} \end{array}$$(6)$$\begin{array}{c} h_{t}=o_{t}*tanh\left(c_{t}\right) \end{array}$$
where *W* and *b* are weight matrices and bias vectors, respectively. σ is the sigmoid activation function and tanh the tanh activation function.

The overall design takes advantage of stacked LSTM architectures, as shown in Figure 3 on the individual branch. After a sequence of frames (a 30 frame window was chosen, which is discussed in experiment A) has been loaded, its geometric and non-geometric features are extracted and fed into two branches with the same structure: a fully connected layer, followed by two LSTM layers and a fully connected layer for the final frame. The output is fed to a softmax layer to obtain the probability for each action class.

#### 3.2.3. Score Fusion

For every skeleton sequence, two sets of feature vectors, a combined static and motion feature vector and the geometric feature vector discussed already, are extracted for every frame to constitute two feature sequences. They are then fed into the two branches of the network to obtain two classification scores. The scores from the two branches are fused, and the maximum score in the resultant vector is assigned as the probability of the recognised action class. The common score fusion methods available are maximise, average and element-wise multiplication. Compared to max-score fusion and average-score fusion, multiply-score fusion has been reported to perform better [49]. Therefore, an element-wise multiplication of the score vectors is performed, followed by a softmax operation to obtain the class with highest probability. Further, when the performance in terms of accuracy varies widely on branches A and B, a threshold is used to either choose from branch A alone or fuse the results from the two.

#### 3.2.4. Application and Logical Layer

To help identify crisis behaviours preceding suicide, the suicide prevention experts in the group conducted a human coding study, and their observations were used to direct supervised learning of the action recognition model. In the human coding study, both incident videos associated with suicide attempts and normal, routine footage were analysed, and eight actions were identified that are typical of normal and crisis behaviours: crouching, leaning on a fence (’Lean’ for short), leaning with head down (’LeanHead’ for short), taking a photo (’Photo’ for short), placing an item on the ground (’Place’ for short), sitting, standing and walking. Of these, Crouch, LeanHead and Place were found to be typical crisis indicators. We trained the action recognition module to recognise them. Furthermore, while such unit actions are an important indicator of suicide intentions, the human coding study discovered that certain behaviour patterns are also high risk indicators. For example, in one crisis case, an individual sat or leaned on the fence for an extended period of time, or walked back and forth from the fence with a combination of sitting, standing, walking and leaning. Therefore, two crisis behaviour patterns should be detected: standing or leaning for a prolonged period of time, and repetitive actions within a given duration.

The logical layer was designed to compute all the above three types of crisis behaviour indicators: unit actions; sit, standing/leaning (since these two actions may shift from one to another without any intentional changes) over a pre-defined duration with a threshold; and repetitive behaviour patterns to check if more than two kinds of actions occur more than a pre-defined number of times at separate timestamps during a given period of time. Such sensitive behaviour patterns include at least one action of ‘sit’ or ‘walk’ to detect patterns of ‘walk back and forth’ or ‘sit and stand up’ shift, and avoid counting the ‘stand/lean’ shift. The logical layer consists of frequency counts of occurrence of actions, and logical rules to detect specific behaviours using recognised actions and computed frequencies. Although the actions are recognised at the frame level, the action count mechanism in the logical layer combines consecutive timestamps with the same label (including void frames in between) to form an action tube with a single label and start and end time stamps, and then counts a single action just once. Furthermore, an action graph for the given period is plotted by the logical layer, and an example action graph is shown in Figure 4, in which the risky behaviour ‘Crouch’ has been captured by the logical layer.

### 3.3. Implementation Details

The action recognition module contains a simple two-layer LSTM component. While previous modules used pretrained weights, the two-branch action recognition model was trained on different datasets from scratch. The network was optimised using Adam and uses a batch size of 32. Half-the-learning rate adjustment strategy was used for certain epochs, and slightly different initial learning rates for different datasets and branches. Common training techniques such as drop-out to counteract overfitting and weighted cross-entropy for unbalanced datasets also used. During training, the two types of skeleton feature sequences were extracted for every 30 frames, and training took place. On a 60,000-frame private training dataset for suicide prevention, the feature extraction and training process took 81 min to complete. Pytorch was the main deep learning platform, and an NVIDIA TITAN X GPU card was used to run the experiments.

During inference, the action class of each pedestrian on each frame is calculated by using it as a mid-key frame and extracting skeleton features from its neighbouring frames (previous 14 and next 15, 30 in total) to constitute a pair of feature sequences for two branches. Then, the sequence pair is fed into the model for action classification. Such an approach enables frame-wise action prediction.

The processing speed for the whole pipeline at resolution 704 by 480 is currently 20 frames/second on average with detection, tracking, pose estimation and action recognition of every frame. Performance increases to 24 frames/second when turning off action recognition, and drops when the number of persons in the scene increases. Since the focus of this study is on recognising single visitors’ actions, building a grouping mechanism into the tracking module will help exclude group visitors. The density of prediction could also be decreased when necessary.

## 4. Experiments

Four sets of experiments were conducted in this study. In the first two experiments, the focus was on testing different features coupled with different network designs for the action recognition model in order to find the optimal choice. The JHMDB dataset [28] was chosen for evaluation for several reasons: firstly, it provides various short unit actions for classification, which are similar to the key scenario here; secondly, it provides 2D ground truth pose information, which can be used to test the performance of skeleton-based action recognition models individually, eliminating the differences caused by different pose estimation algorithms. JHMDB consists of 928 videos for 21 action classes.

In the last two experiments, a private dataset collected from CCTV footage recorded at a location frequently used for suicides in Australia was used (the site is not named in line with best practice for suicide prevention reporting). The site includes public parkland with cliffs adjacent to the ocean. Emergency service callouts indicate that there are approximately 10 completed suicides and 100 emergency call-outs per year. The site is covered by multiple CCTV cameras. A subset of cameras covers defined regions of interest, and the experiments were conducted on videos collected at one of these specific regions. Due to the regulation of and restricted access to the actual suicide attempt videos, the experimental dataset constitutes real normal footage collected on a daily basis, and acted footage that simulates both normal and crisis behaviours identified in the human coding study. One experiment was designed to evaluate action recognition accuracy, and another experiment for action detection and crisis behaviour identification on simulated test videos. The aim was to evaluate the proposed pipeline on its ulility for suicide detection and prevention. Here we use the term ‘action’ to refer to unit actions, and the term ‘behaviour’ to include both unit actions and more complex behaviour patterns, such as walking back and forth.

### 4.1. Experiment 1: Features and Network Structures

In this experiment, the aim was to evaluate the performance of the action recognition model with various feature combinations and different network structures. The main branch (branch A) was used as the base model (Figure 5). The JHMDB ground truth dataset was used to evaluate the performance of the action recognition model only, excluding irrelevant factors such as inconsistent joint detection or pedestrian detection between runs.

Firstly, different types of features were tested and their classification accuracies compared. The base model was used, which was trained with a batch size of 32, initial learning rate of 0.0001 and a decreasing-to-half every 30 epochs strategy for learning rate. The results are shown in Table 2. As may be observed, ‘static feature’ achieved the best accuracy (52.53%) among single features, but was outperformed by all the two feature combinations. The combination of static and motion features (62.81%) was best, at a window size of 30 (frames/sequence). Geometric or motion features alone failed to provide good performance.

To find the best way to utilise geometric features together with static-motion features to further improve the accuracy achieved, three different designs were compared: early fusion, mid fusion and late fusion. In the following experiments, the same setup (batch size 32 and half-the-learning rate adjustment every 30 epochs) was used, and the mean results of three runs are reported.

In early fusion, geometric features were directly concatenated with static-motion features, extending the current 96-dimension static-motion feature vector to 158 by adding the 62 geometric features. The feature vector was fed into the base model described previously (see Figure 5). An initial learning rate of 0.0002 was used to train the network for 97 epochs before overfitting started to occur.

In mid-fusion, non-geometric and geometric features were separately fed into the two branches (see Figure 6). The main difference between this approach and the final design (Figure 3) is that after the LSTM layer, the two sets of output were fused in a fully connected layer of size (hiddensize×2,hiddensize); the input was the concatenated output from the two LSTM layers. It was followed by the last fully connected layer with size (hidden size, number of classes). An initial learning rate of 0.0002 was used to train the network for 35 epochs before overfitting started to occur.

In late fusion, the same base model was trained on static+motion concatenation (non-geometric features), as in the previous experiment (shown in Table 2). Then, another branch was added that was dedicated to geometric features, namely, branch B, for which the same 30 frame sequence length was used, and the 62-dimensional geometric features were extracted from each frame. Thus, tensors of size (batch-size, 30, 62) were fed into this branch for training. An initial learning rate of 0.0003 was used for branch B, and the network was trained for 55 epochs before overfitting started to occur. Models of the two branches were trained separately, and at inference time, the two sets of predictions were combined by multiplying the scores. The performances of these three networks are compared in Table 3.

As may be observed from Table 3, direct concatenation of geometric features with non-geometric static-motion features in early fusion failed to improve the accuracy; rather, it made it worse. This may have been due to the limited capability of LSTM to find the correlations between geometric and non-geometric features. Concatenation may only increase the noise in the data. Mid fusion performed better than early fusion, but was still not able to outperform the single-branch base model with only non-geometric features, as shown in Table 2, 4th row. Using geometric features in a second branch and performing late score fusion achieved the best result (accuracy of 64.01%). Therefore, the late fusion strategy was adopted in the final implementation.

### 4.2. Experiment 2: JHMDB Evaluation

In this experiment, the overall performance of the pipeline was tested in terms of action recognition accuracy on the JHMDB dataset. Instead of using ground truth pose information, the pipeline used its own pedestrian detection, tracking and pose estimation and the final skeleton-based action recognition modules to classify 21 classes of actions. The dataset JHMDB has three partitions, and the pipeline was run on each partition for three runs. The average result over the three partitions is reported. During the training process, the same setup was used, with batch size 32 and half-the-learning rate adjustment every 30 epochs. The results are compared with skeleton-based methods in the literature; see Table 4. As can be observed, the proposed pipeline achieves good performance compared with other work in the literature. In Figure 7, the confusion matrix for the 21 action classes on JHMDB-1 is shown. As can be observed from the matrix, our model shows good discriminating capability for classes such as ‘catch’, ‘clap’, ‘golf’, ‘pick’ and ‘pullup’, but lower accuracies for classes such as ‘climb stairs’, ‘jump’, and ‘kick ball’. We believe the main reason is that our model is good at discriminating unique skeleton movement, but has lower performance for actions with similar body movements; e.g., ‘climb stairs’ is often mistaken for ‘walk’ and ‘kick ball’ is often mistaken for ‘run’. Furthermore, we observe that our model performs generally better on actions with either large or repetitive movements (static posture is seen as repetitive too), whereas small, swift, single-time actions such as ‘sit’ and ‘stand’ are not well-recognised. In these clips, distant frames still contain similar key point positions which are difficult to differentiate; and compared with other actions, these actions do not have sufficient training samples available.

### 4.3. Experiment 3: Action Recognition Model for Suicide Prevention

Based on the human coding study, eight actions of interest were identified: ‘Crouch’, ‘Lean’, ‘LeanHead’, ‘Photo’, ‘Place’, ‘Sit’, ‘Stand’ and ‘Walk’. However, one of the challenges in training the model to identify these behaviours is that some of these behaviours are rare, as crisis events are statistically relatively rare. Furthermore, these events often occur at night, further reducing the number of events within the available RGB footage. To address this, one of the team members acted out all eight actions to simulate both normal and abnormal actions presented by pedestrians, which were used as training samples. In addition, the training set was supplemented with a few clips from normal footage collected on a weekly basis for classes ‘Lean’, ‘Photo’ and ‘Walk’. For each action class, no less than 10 short clips, each one less than 1 min long, were recorded and collected. All the clips were trimmed to contain a single action. The entire training–validation dataset consists of 95 labelled clips and a total of 56,190 frames with human skeleton information. Details can be found in Table 5. A separate set of 15 test clips were recorded with a mixture of both normal and crisis behaviours identified in the human coding study, to test the performance of the visual behaviour analysis pipeline in crisis detection. All videos have a resolution of 704 by 480 at 30 FPS.

As may be observed in Table 5, the data distribution is unbalanced at the frame level. Therefore, the original cross-entropy loss was replaced by weighted cross-entropy loss. Like JHMDB, the dataset was divided by three partitions to eliminate randomness in the experimental setup; each contained approximately 70% of clips for training and 30% for testing. During the actual training process, skeleton feature sequences were first generated for training. To do so, the finetuned YOLOv5-based pedestrian detector was used to detect each pedestrian; then, the detected pedestrians were tracked over each clip and their skeleton movement recorded for every 1 s (30 frames) time window, forming the skeleton sequences. After that, features were extracted from the skeleton sequences and used as input to the proposed network (Figure 3) for training the action recognition model. The model was trained from scratch with the CCTV dataset. Using data split 1, branch A was trained for 50 epochs at an initial learning rate of 0.000015, and branch B was trained for 70 epochs at an initial learning rate of 0.00002. They used a half-learning rate strategy after 40 or 30 epochs, respectively. After both branches were trained, the final classification results were achieved by score fusion. These are shown in Table 6: the performance evaluation of every action class in the test set. The corresponding confusion matrix is shown in Figure 8.

As may be observed, sensitive actions such as ‘Crouch’ and ‘LeanHead’ were recognised very well. Most of the normal actions, such as ‘Walk’, ‘Stand’, ‘Sit’, ‘Lean’, were recognised fairly well too. Two actions, ‘Photo’ and ‘Place’, were not successfully recognised, possibly due to two main reasons: lack of training data on these two classes, and their resemblance to other actions, such as the similarity between taking a photo and leaning on the fence (0.58) and placing an object and standing (0.54).

Average accuracies of the proposed method on each split are shown in Table 7. It may be observed that both split1 and split3 resulted in over 80% accuracy, whereas split2 generally resulted in poorer accuracy, mainly due to the lower performance of branch B. Therefore, it may be deduced that when the two branches produce largely different accuracies, and especially when branch B performs much worse than branch A, the overall fused result will be worse than branch A alone. An arbitrary accuracy difference threshold, for instance, 20% in this case, can be set to choose between the fused result and branch A alone.

We also evaluated our model against classical action recognition models: SlowFast [51], which represents a high performance method for general action recognition tasks, and PoseConv3D [34], which represents the SOTA in skeleton-based action recognition tasks.

Compared with the Slowfast model, as shown in Figure 9, for critical crisis behaviours such as ‘Crouch’, ‘Lean’ and ‘LeanHead’, our approach attained much better recognition rates. We believe this is because our model uses well-designed skeleton features, such as joint locations, angles and distances, which contain important information on certain human postures/movements. Actions ‘Lean’ and ‘LeanHead’, for instance, are very similar to one another, having only small differences in arm position and angle. Therefore, our skeleton-based action recognition model is able to differentiate one from the another, whereas a general classical method such as SlowFast is not. For certain normal actions, such as ‘Walk’ and ‘Sit’, SlowFast performed better, though our approach provided reasonable results too. For the two actions ‘Photo’ and ‘Place’, both approaches failed to provide a satisfactory result due to limited training data. The comparison between the two networks shows the effectiveness and suitability of our skeleton-based approach for the target application.

We then compared our model with the SOTA skeleton-based action recognition model PoseC3D [34] in the intended application using the private CCTV dataset. We used its PYSKL [52] implementation to achieve its best performance, and the results on the three data partitions are shown in the last row of Table 7. As can be observed, our model achieved performance similar to that of PoseC3D, there being a 2.2% difference on average accuracy. However, we believe that PoseC3D lacks flexibility in continuous action detection, as it is designed to stack the heatmaps of the entire video for the classification task, and with 3D-CNN, the computational burden is large. Our model achieved relatively good performance in the intended application with a much lighter design and implementation.

### 4.4. Experiment 4: Action Detection and Logical Analysis for Suicide Prevention

After the action recognition model had been trained and evaluated, the next step was to run the whole pipeline on the 15 simulated test clips in order to evaluate the overall performance of the pipeline in pre-suicide crisis detection, which can be potentially used for early prevention. These 15 test clips consist of a mixture of behaviours, which are intended to simulate both normal and crisis behaviours found in the human coding study. Of the 15 clips, 10 contain only normal behaviours and the other 5 contain high risk crisis behaviours. The unit action ground truth annotations and list of high risk behaviours with timestamps were available, with the latter including both unit actions and a mixture of behaviour patterns.

When running the pipeline on the 15 test clips, both sensitive and normal actions were recognised, along with detecting and tracking the pedestrians and estimating pose, as shown in Figure 10. Secondly the action recognition results were used by the logical layer to generate temporal action tubes, which were used for evaluation of temporal action detection on a class-wise basis. The average precision (AP) of each action class was computed between ground truth and predicted actions. In Table 8, the temporal action detection evaluation using mAP at different IoU thresholds is shown. The mAP@0.5 for all classes was 32.53. It can be observed that some actions were well detected, such as ‘Crouch’, ‘Sit’, ‘Stand’ and ‘Walk’, which the algorithm showed relatively good performance for in the classification task too (Figure 8). On the other hand, some actions were detected well in classification but not as well in temporal detection, including ‘Lean’ and ‘LeanHead’. The reasons were likely due to the similarity between these two actions and the frequency of shift between them. When they change from one to another for a short period of time, the model might fail to detect them, or only detect a partial duration of each. Two actions, ‘Place’ and ‘Photo’, failed to be detected at all, as they have not been annotated in large enough numbers in the trimmed videos.

Thirdly, the action recognition results were used as input to the logical layer to capture more complex behaviour patterns. Action graphs have been plotted for each short clip. Examples are shown in Figure 11. As shown in the top left, which is from test 11, the following crisis behaviours were identified by the logical layer: in test 11, pedestrian 0.0001 demonstrated risky behaviour with ‘Lean/Stand’ for a long duration of 105.00 s from 18.37 to 123.37 s, ‘Crouch’ at timepoint 123.37 s and ‘Lean/Stand’ for a long duration of 116.33 s from 130.87 to 247.20. In Figure 11, top right, test 14: pedestrian 0.0001 demonstrated risky behaviour—‘LeanHead’ at timepoint 36.13 s, ‘Crouch’ at timepoint 47.80 s, ‘Crouch’ at timepoint 67.77 s and total action counts of: ‘Crouch’: 2, ‘Lean’: 4, ‘LeadHead’: 1, ‘Photo’: 0, ‘Place’: 0, ‘Sit’: 0, ‘Stand’: 2, ‘Walk:’ 2, where the number indicates how often the corresponding action occurs. The ‘long duration’ threshold can be modified to different values, and a value of 100 s was used in the simulated cases.

The overall performance of crisis identification on test clips is shown in Table 9 and Table 10. The former table shows video-level metrics of 80% recall and 86.7% accuracy at the clip level. At a finer level (the latter table), of the 12 occurrences of crisis behaviours, 8 were correctly identified. There were 2 missed occurrences of ‘Place’, 1 of ‘LeanHead’ and 1 of ‘Repetitive Pattern’, as the tracking module mistakenly generated a new ID for an already tracked pedestrian, leading to two sets of actions. A total of 11 crisis behaviours were identified: 8 were true (precision 72.7%), 1 ‘Lean’ was mis-classified as ‘LeadHead’ and two were mis-identified ‘Crouch’ due to a sitting down and standing up action that involved two gestures resembling ‘Crouch’. It can be observed that the video clips with crisis behaviours can be identified with high accuracy and recall, whereas individual crisis behaviour detection still needs further improvement because mis-identification often occurs for similar gestures. Additionally, the performances of all the previous modules, such as pedestrian detection and tracking, can influence the final crisis behaviour identification.

## 5. Discussion

In this work, a new human-centric approach to building an intelligent surveillance visual processing pipeline has been presented, focussing on detecting behaviours which precede a suicide attempt. A modular design is proposed, from pedestrian detection, tracking and pose estimation to action recognition and a logical analysis layer on top to capture crisis behaviour patterns, such as long durations of certain actions or repetitive patterns. Such a design facilitates analysis of results from each component for a variety of purposes. For instance, the tracking output can be used not only as input to the succeeding modules, but also extended for trajectory analysis in a future study. Secondly, a new 2D skeleton-based action recognition model is proposed, which uses two branches to process different features extracted from human skeleton sequences. Their scores are fused to achieve better classification performance. The action recognition model was evaluated on a public dataset, JHMDB, and achieved good performance. Thirdly, the model was trained on a private dataset for recognising unit actions of interest identified by a human coding study. Finally, a logical analysis layer, which is based on knowledge elicited from the human coding study, is used to generate single action tubes from frame-wise action labels, and also capture more complex behaviour patterns. The proposed pipeline is able to detect both normal and high-risk unit actions with mAP@0.5 of 32.53. When taking more complex high risk behaviour patterns into consideration, an average accuracy of 86.7% and 80% recall at the video level, and 72.7% precision and 66.7% recall at the individual crisis behaviour level, have been achieved.

In future work, performance will be further improved by recognising swift actions, such as placing objects on the ground. Secondly, the current tracking module will be used to expand the trajectory analysis API so that trajectory information can be used as a high-risk indicator for suicide prevention. Furthermore, the visual behaviour analysis pipeline will be applied to suicide prevention in different locations and scenarios for more general usage in this domain.

## Figures and Tables

**Figure 1 sensors-22-04488-f001:**
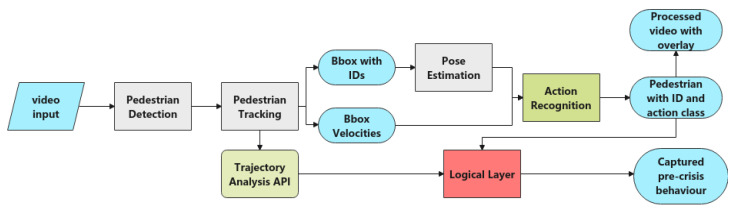
Structure of proposed visual processing pipeline.

**Figure 2 sensors-22-04488-f002:**
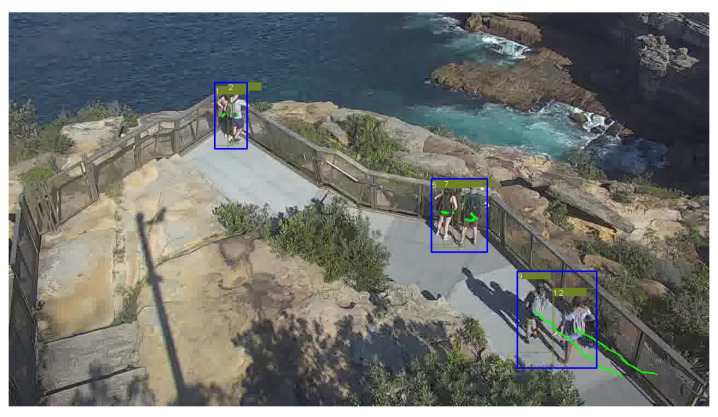
Demonstration of the trajectory analysis API: grouping and trajectories for a duration of 2 s (60 consecutive frames) with trajectories been shown in green.

**Figure 3 sensors-22-04488-f003:**
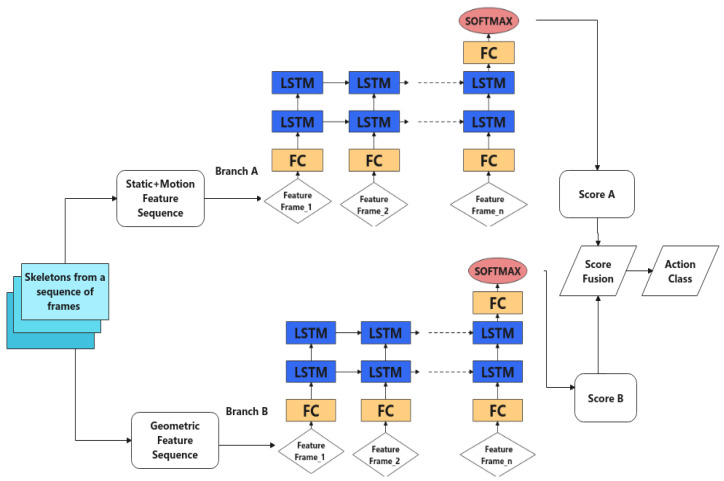
Network structure of proposed action recognition model.

**Figure 4 sensors-22-04488-f004:**
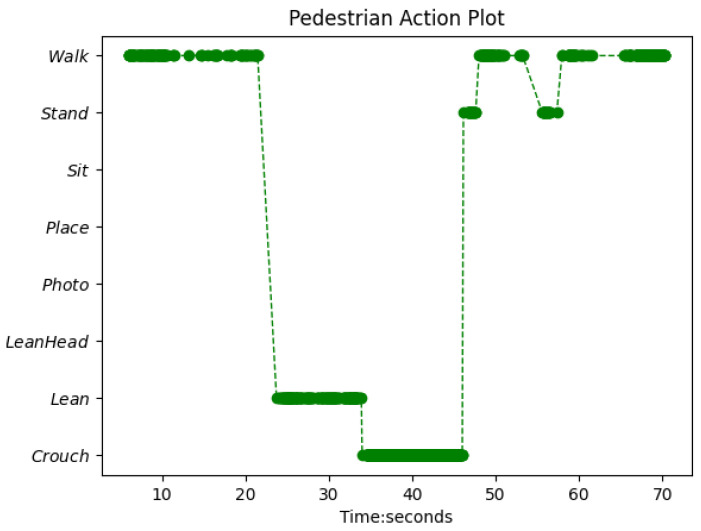
Action graph of test clip 13. As the prediction is frame-wise, there are void time-points in between where the actions at the time-points were not recognised due to low confidence.

**Figure 5 sensors-22-04488-f005:**
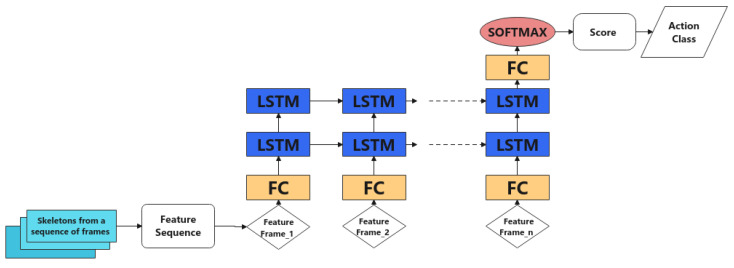
The base model was used as the baseline for evaluating the performance of action recognition on various feature combinations.

**Figure 6 sensors-22-04488-f006:**
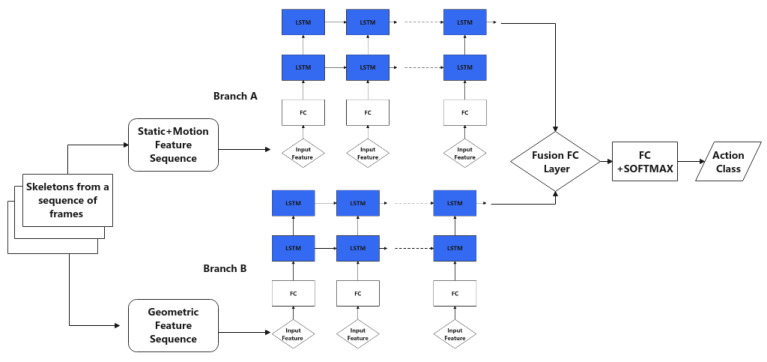
Mid-fusion network structure.

**Figure 7 sensors-22-04488-f007:**
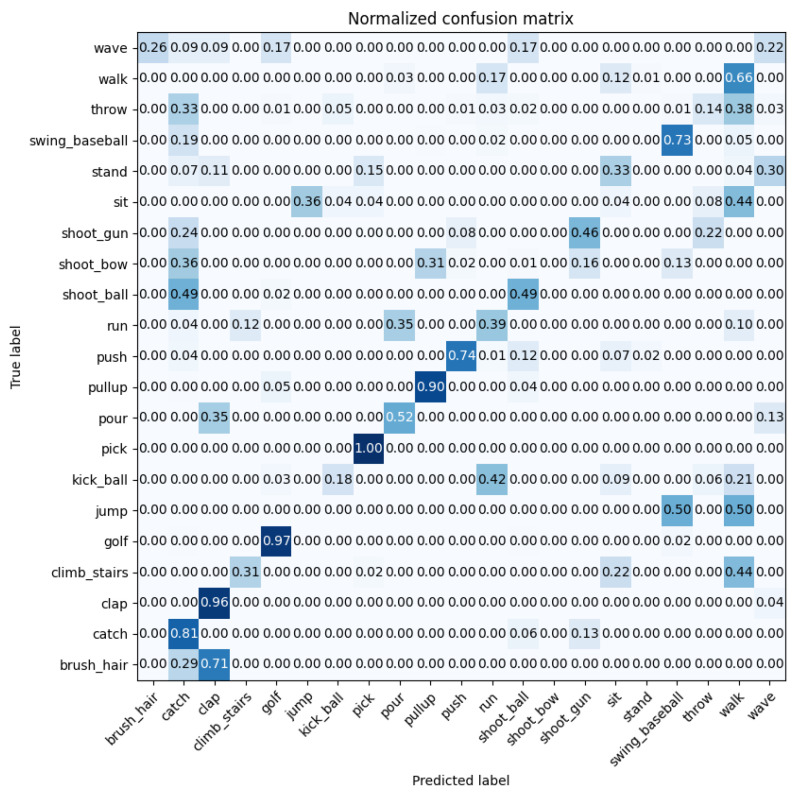
Confusion matrix of JHMDB-split1 obtained by the proposed pipeline, with darker color indicates higher hit.

**Figure 8 sensors-22-04488-f008:**
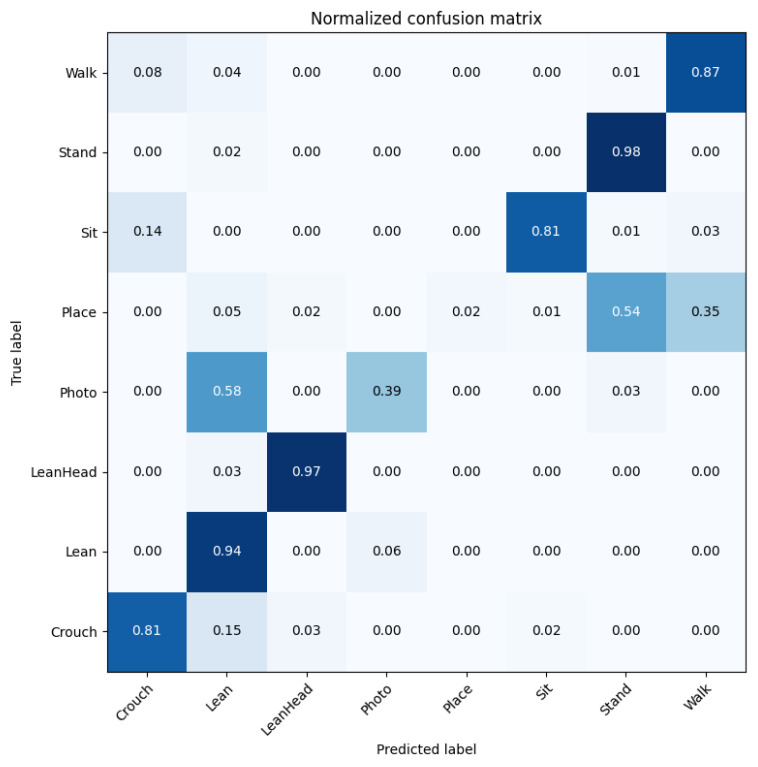
Confusion matrix of action recognition task on CCTV dataset-split1 obtained by the proposed pipeline, with darker color indicates higher hit.

**Figure 9 sensors-22-04488-f009:**
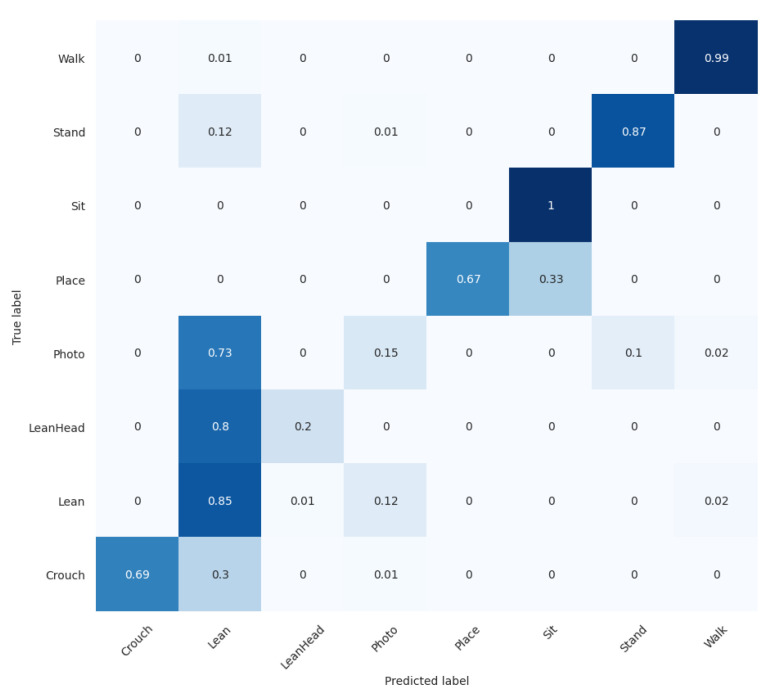
Confusion matrix of the action recognition task on CCTV dataset-split1 performed by the SlowFast network, with darker color indicates higher hit.

**Figure 10 sensors-22-04488-f010:**
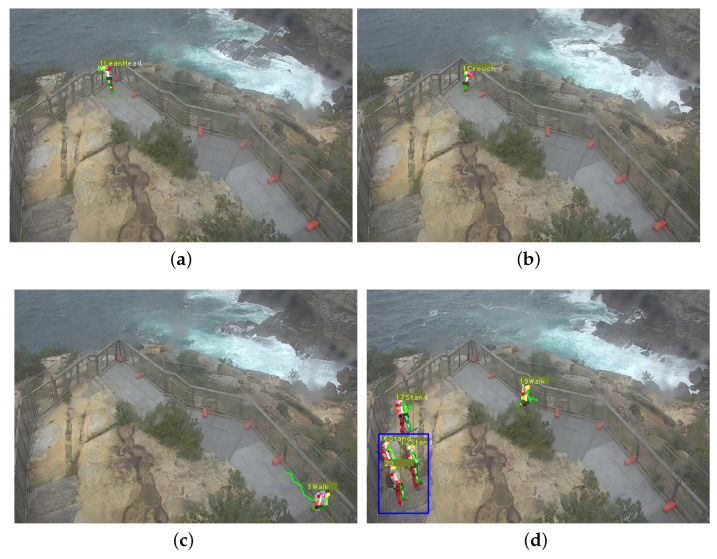
Examples of recognition results on the CCTV testing clips: in (**a**) a pedestrian with “LeanHead” action has been recognised, similarly in (**b**) “Crouch”, in (**c**) “Walk”, and (**d**) is a multiple-pedestrians scene with both “Stand” and “Walk” actions been recognised.

**Figure 11 sensors-22-04488-f011:**
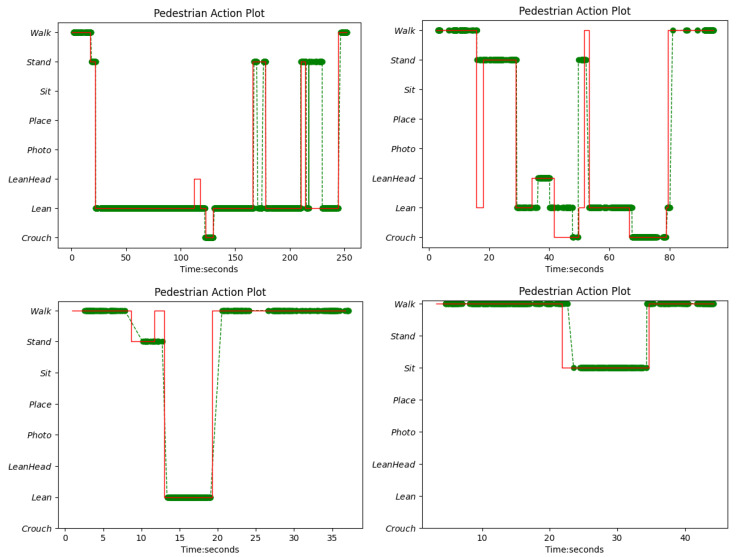
Sample high-risk action graphs (**top left** and **top right**) and normal-action graphs (**bottom left** and **bottom right**) with ground truth. The green dotted line is the prediction and the red line is the ground truth.

**Table 1 sensors-22-04488-t001:** Geometric features.

Geometric Feature List	Dimension	Description
Limb Orientation	13	Limb angle
Limb Length	13	Limb lengths
Inner Angles	13	For every 3 adjacent joints, calculate the inner angle between 2 limbs that share the same joint
Long Length	13	For every 3 adjacent joints, calculate the distance between 2 non-adjacent joints
End Length	10	For every end joint (defined as joint that has only one side connected to another joint, including nose, wrists and ankles), calculate the distance between every two end joints
Total	62	

**Table 2 sensors-22-04488-t002:** Evaluation of JHMDB-1-GT with different types of feature combinations.

Feature Type	Sequence Length	Dimension	Accuracy (%)
Static	30	28	52.53
Motion	30	68	38.79
Geometric	30	62	41.91
Static + Motion concatenation	30	96	62.81
Static + Motion concatenation	15	96	58.10
Static + Motion concatenation	45	96	61.39

**Table 3 sensors-22-04488-t003:** Results on JHMDB-1-GT with 3 types of model structures and feature inputs.

Model Structure	Feature Dimension	Accuracy (%)
Early Fusion	158	45.28
Mid Fusion	[96, 62]	54.51
Late Fusion: branch A	96	62.81
Late Fusion: branch B	62	43.06
* Late Fusion: score fusion	[96, 62]	64.01

* The final adopted structure.

**Table 4 sensors-22-04488-t004:** Results (accuracy, %) on JHMDB compared with other skeleton-based methods.

Method	JHMDB-1	JHMDB-1-GT	JHMDB
Zholfaghari et al. [50]	45.5	56.8	N/A
PoTion [33]	59.1	67.9	57.0
DD-Net [31]	N/A	77.2	N/A
EHPI [35]	60.3	65.5	60.5
Proposed method	61.24	64.01	60.2

**Table 5 sensors-22-04488-t005:** Data distribution of 8 action classes in the private CCTV training dataset.

Data Types	Crouch	Lean	Lean-Head	Photo	Place	Sit	Stand	Walk
Num of Clips	10	14	10	10	10	15	14	12
Num of Frames	7163	9894	6534	1455	1045	14.451	11.944	3704

**Table 6 sensors-22-04488-t006:** Evaluation of action recognition on private CCTV dataset-split1.

Actions	Precision	Recall	F1-Score	Support
Crouch	0.7385	0.8125	0.7737	2165
Lean	0.7593	0.9318	0.8367	2434
LeanHead	0.9621	0.9689	0.9655	1545
Photo	0.5016	0.3883	0.4378	394
Place	0.8571	0.0245	0.0476	245
Sit	0.9920	0.8208	0.8983	3912
Stand	0.9282	0.9844	0.9555	3021
Average	0.8198	0.7045	0.7022	1959

**Table 7 sensors-22-04488-t007:** Evaluation (accuracy, %) on CCTV the dataset—3 partitions.

Accuracy	Split1	Split2	Split3	Average
Branch A	84.21	78.01	82.95	81.72
Branch B	74.60	48.56	67.21	63.45
Fused	86.52	75.62	84.22	82.12
PoseC3D	89.71	81.25	82.21	84.39

**Table 8 sensors-22-04488-t008:** Class-wise temporal action detection results on the test dataset with mAP at different IoU thresholds α.

IoU	Crouch	Lean	Lean-Head	Photo	Place	Sit	Stand	Walk
0.1	100	46.7	33.3	0	0	66.7	53.5	64.8
0.2	100	46.7	33.3	0	0	66.7	53.5	60.2
0.3	62.5	43.3	33.3	0	0	66.7	53.5	60.2
0.4	62.5	38.7	33.3	0	0	66.7	35.0	49.6
0.5	62.5	30.6	33.3	0	0	66.7	25.7	41.4
0.6	62.5	30.6	0	0	0	66.7	22.2	30.9
0.7	62.5	22.7	0	0	0	66.7	5.5	20.2
0.8	37.5	14.0	0	0	0	66.7	2.4	16.3
0.9	25.0	7.7	0	0	0	0	0	3
1	0	0	0	0	0	0	0	0

**Table 9 sensors-22-04488-t009:** Evaluation of crisis video identification on test clips.

Total Clips	TP	FP	TN	FN
15	4	1	9	1

**Table 10 sensors-22-04488-t010:** Evaluation of crisis behaviour identification on test clips.

Total Crisis Behaviours	Total Identification	Correct	False	Miss
12	11	8	3	4

## Data Availability

Not applicable.

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
