# Peer review of "Towards Building a Visual Behaviour Analysis Pipeline for Suicide Detection and Prevention"

_sensors, 2022, doi:10.3390/s22124488_

Round 1

Reviewer 1 Report

The paper presents a new human-centric approach to building an intelligent surveillance visual processing pipeline has been presented, focussing on detecting behaviours which precede a suicide attempt. It is a topic of interest to the researchers in the related areas. For the reader, however, a number of points need clarifying and certain statements require further justification. My detailed comments are as follows:

(1) Suicide prevention researchers will judge and interfere with people still in the suicide latency period, while this paper focuses on suicide behavior based on the human skeletal framework. Is there a chance to intervene and stop suicidal behavior when it is identified? Please the authors once again reconsider the research background and objectives.

(2)The author design to study a new model needs to consider its practical application environment. In the article, readers cannot clearly understand the population audience suitable for the model. I hope the author can make it clear.

(3)This paper evaluates the performance of a new model using a small sample dataset, and the reliability of the data results remains to be verified. Whether the authors can remedy this problem by other means to make the model more convincing.

(4)It is noteworthy that your paper requires careful editing of the format.

Author Response

The authors thank the reviewers for their thorough reading and insightful comments on the paper. The paper has been revised based on the feedback, and we address the reviewers’ comments below:

(1) Suicide prevention researchers will judge and interfere with people still in the suicide latency period, while this paper focuses on suicide behavior based on the human skeletal framework. Is there a chance to intervene and stop suicidal behavior when it is identified? Please the authors once again reconsider the research background and objectives.

In this research, potential pre-crisis behaviours were carefully analysed and determined by suicide prevention experts through a human coding study [10, 12]. The important action cues are the ones which indicate suicide attention, such as leaning with head down which is a sign of depression, and walking repetitively around the same area which indicates a sign of hesitation. Such behaviours can be spotted hours before the actual attempt, which allows sufficient time for intervention. Our paper focuses on these behaviour cues in the buildup period towards a potential suicide attempt. The actual crisis actions like climbing the fence are already too late for prevention and not of interest to suicide prevention experts. We therefore ignore such actions. See changes on page 2.

[10] Mishara, B.L.; Bardon, C.; Dupont, S. Can CCTV identify people in public transit stations who are at risk of attempting suicide? An analysis of CCTV video recordings of attempters and a comparative investigation. BMC public health 2016, 16, 1–10.

[12] Onie, S.; Li, X.; Liang, M.; Sowmya, A.; Larsen, M.E. The Use of Closed-Circuit Television and Video in Suicide Prevention: 756 Narrative Review and Future Directions. JMIR Ment Health 2021, 8, e2766

(2)The author design to study a new model needs to consider its practical application environment. In the article, readers cannot clearly understand the population audience suitable for the model. I hope the author can make it clear.

Based on the key findings that human body gestures can be used for the detection of social signals that potentially indicate intention preceding a suicide attempt [11], a new 2D skeleton-based action recognition algorithm has been proposed. Suicide related datasets are usually limited in both data size and accessibility. Using recognition based on bone points , combined with a high-quality attitude detector, skeleton-based approaches can often achieve good recognition in the case of sparse data and have strong generalization ability [CVPR 2022 oral]. Therefore we believe that our skeleton-based model is a suitable choice for a practical application environment, and our experiments prove its effectiveness as well.

[11]Reid, S.; Coleman, S.; Kerr, D.; Vance, P.; O’Neill, S. Feature Extraction with Computational Intelligence for Head Pose Estimation. In Proceedings of the 2018 IEEE Symposium Series on Computational Intelligence (SSCI), 2018, pp. 1269–1274.

[CVPR 2022 oral] poseconv3d open source: a new paradigm of action recognition based on human posture

(3)This paper evaluates the performance of a new model using a small sample dataset, and the reliability of the data results remains to be verified. Whether the authors can remedy this problem by other means to make the model more convincing.

To compare our model’s performance with SOTA algorithms, we have added one further experiment from PoseC3D [34] on the intended application using this private dataset. We use its PYSKL[53] implementation to achieve its best performance and its results on the 3 splits are now shown in Table 7. As can be observed, our model achieves performance close to PoseC3D, with 2.2% difference in average accuracy. However we believe PoseC3D lacks flexibility on continuous action detection as it is designed to stack the heatmaps of the entire video for the classification task, and with 3D-CNN the computational burden is large. Our model achieves relatively good performance on the intended application with a much lighter design and implementation. See changes on page 17.

[34]Duan, H., Zhao, Y., Chen, K., Shao, D., Lin, D., & Dai, B. (2021). Revisiting skeleton-based action recognition. arXiv preprint arXiv:2104.13586.

[53] Duan, H., Wang, J., Chen, K., & Lin, D. (2022). PYSKL: Towards Good Practices for Skeleton Action Recognition. arXiv preprint arXiv:2205.09443.

(4)It is noteworthy that your paper requires careful editing of the format.

Minor changes have been made on the format. The paper has been thoroughly proof read by a native English speaker (co-author ML) and other co-authors who are fluent in Englsh (AS, SO).

Reviewer 2 Report

The authors proposed a method for detection of behaviors potentially leading to suicide, which is a very important topic. Although the proposed feature vector is simple and there are is not much novelty in each module of the recognition pipeline, the algorithm as a whole is sound and can be considered as novel. The authors have made a solid review of the literature, not only on the subject of automatic action recognition, human detection and tracking but also concerning pre-suicidal behaviors.  Making use of information published in public health-related journals helped to create a method that has a chance to be used in practical applications.

The accuracy of the method is far from perfect. It outperforms other methods in one of three cases. It is therefore not a complete solution to the given problem. However, I think, the method is a solid base for future work and may help to develop a high-performance suicidal prevention system.

In my opinion, the paper is suitable for publication. I only have a few very minor comments.

1.      Please consider uploading Fig. 3 in higher resolution or at least enlarge "Feature frame" blocks because this caption is almost illegible.

2.      In table 6 please add a row with average statistics for all actions.

3.      I detected one typo: "which are are strong at extracting visual features" - double "are".

Author Response

The authors thank the reviewers for their thorough reading and insightful comments on the paper. The paper has been revised based on the feedback, and we address reviewers’ comments below.

1.      Please consider uploading Fig. 3 in higher resolution or at least enlarge "Feature frame" blocks because this caption is almost illegible.

The resolution of Figure 3 has been modified to a higher resolution.

2.      In table 6 please add a row with average statistics for all actions.

An average row has been added to the end of Table 6.

3.      I detected one typo: "which are are strong at extracting visual features" - double "are".

Thank you for the sharp eye. The mistake has been corrected.